Linking a genetic defect in migraine to spreading depression in a computational model

Dahlem Markus A. 1 markus.dahlem@gmail.com
Schumacher Julia 2
Hübel Niklas 3
1 Department of Physics, Humboldt Universität zu Berlin , Berlin , Germany
2 Bernstein Center for Computational Neuroscience, Technische Universität Berlin , Berlin , Germany
3 Department of Theoretical Physics, Technische Universität Berlin , Berlin , Germany
Sensi Stefano
Electronic publication date: 2014 May 8
Publication date: 2014
Volume: 2
Electronic Location ID: e379
Received 2014 Mar 27; Accepted 2014 Apr 23
Copyright: © 2014 Dahlem et al.
Copyright year: 2014
Copyright holder: Dahlem et al.
License: This is an open access article distributed under the terms of the Creative Commons Attribution License, which permits unrestricted use, distribution, reproduction and adaptation in any medium and for any purpose provided that it is properly attributed. For attribution, the original author(s), title, publication source (PeerJ) and either DOI or URL of the article must be cited.
License URL: https://creativecommons.org/licenses/by/4.0/

Keywords: Inactivation, Hyperexcitability, Hypoexcitability, Familial hemiplegic migraine, Hodgkin–Huxley model, Potassium channel, Action potential, Threshold, Anoxia, Ions

Funding: Bundesministerium für Bildung und Forschung BMBF 01GQ1001A 01GQ1001B 01GQ1109 This work was supported by the Bundesministerium für Bildung und Forschung (BMBF 01GQ1001B, 01GQ1109) within the Bernstein Center of Computational Neuroscience Berlin. The funders had no role in study design, data collection and analysis, decision to publish, or preparation of the manuscript.

==============================
Familial hemiplegic migraine (FHM) is a rare subtype of migraine with aura. A mutation causing FHM type 3 (FHM3) has been identified in SCN1A encoding the Nav1.1 Na+ channel. This genetic defect affects the inactivation gate. While the Na+ tail currents following voltage steps are consistent with both hyperexcitability and hypoexcitability, in this computational study, we investigate functional consequences beyond these isolated events. Our extended Hodgkin–Huxley framework establishes a connection between genotype and cellular phenotype, i.e., the pathophysiological dynamics that spans over multiple time scales and is relevant to migraine with aura. In particular, we investigate the dynamical repertoire from normal spiking (milliseconds) to spreading depression and anoxic depolarization (tens of seconds) and show that FHM3 mutations render gray matter tissue more vulnerable to spreading depression despite opposing effects associated with action potential generation. We conclude that the classification in terms of hypoexcitability vs. hyperexcitability is too simple a scheme. Our mathematical analysis provides further basic insight into also previously discussed criticisms against this scheme based on psychophysical and clinical data.

Introduction

Familial hemiplegic migraine (FHM) is a rare monogenic, autosomal dominantly inherited syndrome with hemiparesis during the aura phase of migraine. Three distinct genetic mutations for FHM have been identified, in the CACNA1A calcium channel gene (FHM1), in the ATP1A2 Na,K-ATPase gene (FHM2), and in the SCN1A sodium channel gene (FHM3). It has been proposed that all three phenotypes reflect hyperexcitability in the form of increased susceptibility for spreading depression (SD) (van den Maagdenberg et al., 2007; Pietrobon, 2010). However, the functional connection between the molecular findings and a facilitated generation of SD is unclear.

To determine the electrophysiological consequences of such a genetic defect, we integrate a mutation of FHM3 into three types of computational models of neuronal dynamics. This allows us to bridge the gap between genotype and phenotype. A similar approach was used by Clancy & Rudy (1999). We use a standard Hodgkin–Huxley model for action potentials (AP) (Hodgkin & Huxley, 1952) and a model of SD (Hübel, Schöll & Dahlem, 2014) to evaluate the change in the threshold of generating SD by tolerating various brief intervals of transient ischemic attacks. Moreover, we use a model for anoxic depolarization (AD) (Zandt et al., 2011) that is derived from a seizure model (Cressman et al., 2009; Cressman et al., 2011) as a test of the robustness of our results.

The paper is organized as follows. In the Methods we introduce three computational models and our method to incorporate measured tail currents in FHM3 (Dichgans et al., 2005; Vanmolkot et al., 2007) into the Hodgkin–Huxley framework. In the Results we present simulations and analysis of the wild-type and mutant models. We end with the Discussion where we focus on three topics: (i) the appropriateness of the terms hypoexcitable vs. hyperexcitable, (ii) the seemingly paradoxically increased susceptibility to SD in the mutant model if one considers the firing rate, a measure that is usually used to quantify slow neural dynamics, and (iii) the inadequate concept of a threshold as a quantity measured by a single value.

Methods

All three models are based on Hodgkin–Huxley type dynamics with different degree of complexity from the classical model to a second generation with time-dependent ion concentrations.

Hodgkin–Huxley model

The Hodgkin–Huxley (HH) model is one of the most widely used computational models in neuroscience. It is a conductance-based neuron model (Hodgkin & Huxley, 1952) and consists of four differential equations describing the membrane potential V and three gating variables m, n and h that determine the conductances of potassium and sodium channels. The change in membrane potential is proportional to the current that is flowing across the membrane with the proportionality constant given by the capacitance of the membrane Cm. The individual currents are modeled as the conductance gi of the respective channel times the driving force, which is given by the difference between the membrane potential and the respective ion’s reversal potential Ei, where i∈{K, Na, leak}. Note that the conductance gj for voltage-gated channels, i.e., j∈{K, Na}, is given by the maximal conductance g¯j times the respective gating variables as introduced below. The model takes into account a sodium current INa+, a potassium current IK+, a leak current Ileak that is carried by unspecified ions, and an applied current Iapp. (1) dVdt=−1CmINa++IK++Ileak−Iapp,

(2) INa+=g¯Nam3hV−ENa,

(3) IK+=g¯Kn4V−EK,

(4) Ileak=glV−Eleak.

In the HH model the potassium current is modeled as a delayed rectifier current with activation gate n while the sodium current is described by a transient current with an activation gate m and an inactivation gate h. All gating variables are voltage dependent and are given by the following equations: (5) dxdt=x∞−xτxwith

(6) x∞=αxαx+βxand

(7) τx=1αx+βxforx∈n,m,h.

x∞ describes the steady-state of the gating variables and τx is the time constant.

Table 1 Model parameters for the Hodgkin–Huxley model.

Name	Value & unit	Description	
Cm	1 µF/cm2	Membrane capacitance	
g¯Na	120 m/cm2	Max. sodium conductance	
g¯K	36 m/cm2	Max. potassium leak conductance	
gl	0.3 m/cm2	Leak conductance	
E Na	50 mV	Sodium reversal potential	
E K	−77 mV	Potassium reversal potential	
E leak	−54.402 mV	Leak reversal potential	

The rate equations for αx and βx are voltage-dependent and given by (8) αm=0.1V+401−exp−V+40/10,

(9) βm=4exp−V+65/18,

(10) αn=0.01V+551−exp−V+55/10,

(11) βn=0.125exp−V+65/80,

(12) αh=0.07exp−V+65/20,

(13) βh=11+exp−0.1V+35.

This model is capable of producing action potentials in response to depolarizations of the membrane caused by an appropriate externally applied current Iapp. All model parameters that were used in the simulations of the HH model can be found in Table 1. It is interesting to remark that trying to study the effect of the mutation in a reduced two-dimensional model in the phase plane did not lead to promising results because the mutation quickly led to bistability, which is consistant with our results of a prolonged plateau of action potential and early depolarization block in the form of bistability.

Spreading depression model

The classical HH model neglects the time-dependency of ion concentrations caused by spiking dynamics. Ions accumulate very slowly but also progressively due to the fluxes across the neuronal membrane. Therefore, changes in concentrations become significant either in the course of many rapid action potentials or under metabolic stress with insufficient ion pump activity, such as during transient ischemic attacks. Hence both the onset of spiking and also the response to reduced ion pump activity are of interest. These can be modeled by the spreading depression model described in more detail by Hübel, Schöll & Dahlem (2014).

This model is also based on HH dynamics, but uses several changes and extensions. Instead of an unspecified leak current, a combined Na+–K+-leak current is used. The equations for sodium and potassium currents, including a pump current Ip that is introduced below, therefore change to (14) INa+=gNal+g¯Nagm3h⋅V−ENa+3Ip,

(15) IK+=gKl+g¯Kgn4⋅V−EK−2Ip.

Furthermore, the SD model uses dynamic ion concentrations to be able to model the breakdown of the ion gradients that is observed during SD. The intracellular potassium concentration Ki and extracellular potassium concentration Ke are modeled explicitly as dynamical variables, while the intra- and extracellular sodium concentrations (Nai and Nae) are computed from the potassium concentration due to the constraint of electroneutrality (16) dKidt=−γωiIK+,

(17) dKedt=γωeIK++JdiffKe

(18) Nai=Nai0−Ki+Ki0,

(19) Nae=ωiωeNai0−Nai+Nae0.

The factor γ converts currents to ion fluxes and depends on the membrane surface Am and Faraday’s constant F: (20) γ=AmF,

ωi and ωe are constants describing the intra- and extracellular volume, respectively, and the buffer flux Jdiff is (21) Jdiff=FdiffKbath−Ke.

An overview of all constants and the values that were used in the simulations can be found in Table 2.

Table 2 Model parameters for the SD model.

Name	Value & unit	Description	
Cm	1 µF/cm2	Membrane capacitance	
gNal	0.0175 m/cm2	Sodium leak conductance	
gNag	100 m/cm2	Max. gated sodium conductance	
gKl	0.05 m/cm2	Potassium leak conductance	
gKg	40 m/cm2	Max. gated potassium conductance	
Nai	27 mM/l	ECS sodium concentration	
Nae	120 mM/l	ICS sodium concentration	
Ki	130.99 mM/l	ECS potassium concentration	
Ke	4 mM/l	ICS potassium concentration	
E Na	39.74 mV	Sodium reversal potential	
E K	−92.94 mV	Potassium reversal potential	
ωi	2,160 µm3	Volume of ICS	
ωe	720 µm3	Volume of ECS	
F	96,485 C/Mol	Faraday’s constant	
Am	922 µm2	Membrane surface	
γ	9.556e−6 µm2MolC	Conversion factor	
ρ	5.25 µA/cm2	Max. pump current	
ϕ	3/ms	Gating timescale parameter	
F diff	3.75e−5/ms	Diffusion parameter	
Kbath	4 mM/l	Potassium bath concentration	

If ion concentrations are time-dependent, they actually change drastically during neuronal activity. To still maintain homeostasis an ion pump has to be included that pumps Na+ ions out of and K+ ions into the cell at a 3/2 ratio. The pump current thus depends on the extracellular potassium and the intracellular sodium concentration. The pump is modeled according to Barreto & Cressman (2011) (22) IpNai,Ke=ρ1+exp25−Nai3−11+exp5.5−Ke−1,

with ρ being the pump current strength. Note that the pump current also shows up in the equations for Na+- and K+-currents (Eqs. (14) and (15)).

As a result of the dynamic ion concentrations also the reversal potentials become dynamic (23) Eion=26.64zionlnione/ioni.

The fast gating dynamics of the m-gate is modeled adiabatically as (24) m=m∞V.

Note that in this model shifted versions of the rate equations are used (Cressman et al., 2009; Cressman et al., 2011) (25) αm=0.1V+301−exp−V+30/10,

(26) βm=4exp−V+55/18,

(27) αn=0.01V+341−exp−V+34/10,

(28) βn=0.125exp−V+44/80,

(29) αh=0.07exp−V+44/20,

(30) βh=11+exp−0.1V+14.

Furthermore, the time constants are scaled by a factor ϕ (31) τx=1ϕαx+βx.

In contrast to Hübel, Schöll & Dahlem (2014) we did not reduce the dimension of the model further by assuming a linear or sigmoidal relation between n and h. Instead, h was kept dynamic since the changes caused by the mutation affect the h-gate.

Anoxia model

As a test of the robustness of our results we investigate the effects of FHM3 also in a mutant model of anoxia (Zandt et al., 2011). In fact, migraine with aura has been linked to a higher risk of ischemic stroke (Kurth & Diener, 2012). For furthere details on the rationale, see the Sec. Results.

Table 3 Model parameters for anoxia model.

Name	Value & unit	Description	
Cm	1 µF/cm2	Membrane capacitance	
gNal	0.0175 m/cm2	Sodium leak conductance	
gNag	100 m/cm2	Max. gated sodium conductance	
gKl	0.05 m/cm2	Potassium leak conductance	
gKg	40 m/cm2	Max. gated potassium conductance	
gCll	0.05 m/cm2	Chloride leak conductance	
Nai	27 mM/l	ECS sodium concentration	
Nae	120 mM/l	ICS sodium concentration	
Ki	130.99 mM/l	ECS potassium concentration	
Ke	4 mM/l	ICS potassium concentration	
E Na	39.74 mV	Sodium reversal potential	
E K	−92.94 mV	Potassium reversal potential	
ϕ	3/ms	Gating timescale parameter	
A/VF	0.044 mMs/mAcm2	Conversion factor	
β	2.0	Ratio ICS/ECS	
ρ	28.1 µA/cm2	Na–K-Pump rate	
G	66 mM/s	Glial buffering rate for K+	
ϵ	1.3 s−1	Diffusion rate	
k ∞	4.0 mM	Concentration K+ in blood	
T	310 K	Absolute temperature	

The anoxia model is similar to the SD model, but uses five more dynamic variables, in particular, it also models chloride ion dynamics. The other dimensions are due to explicitly modeling intra- and extracellular ion concentrations and not assuming mass conservation, and also electroneutrality is not assumed in this model.

Therefore, in addition to Na+- and K+-currents as in Eqs. (14) and (15) a chloride (Cl−) channel is included, which contributes to the leak current (32) dVdt=−1CmINa++IK++ICl

(33) ICl−=gCllV−ECl.

Intra- and extracellular ion concentrations are dynamic and modeled as (34) dNaidt=−AVFINa+

(35) dNaedt=βAVFINa+

(36) dClidt=−AVFICl−

(37) dCledt=βAVFICl−

(38) dKidt=−AVFIK+

(39) dKedt=βAVFIK+−Ig−Id.

The same pump current as in the SD model is used (Eq. (22)). While the total amount of sodium and chloride is constant, the extracellular potassium concentration can be buffered by glial cells (Ig) and diffuse into and out of the blood (Id) (40) Ig=G1+exp18−Ke2.5−1,

(41) Id=ϵKe−k∞,

h and n are dynamic and given by Eqs. (5), (6) and (25)–(31). The sodium activation gate m is adiabatically modeled as in Eq. (24). For parameter values see Table 3.

Under physiological conditions this model behaves normally, as it responds with a single action potential to a short current pulse and with periodic firing when a larger current of 1.5 mA/cm2 or more is injected (not shown). This model is also able to show seizure activity (Cressman et al., 2009; Cressman et al., 2011).

Modified time constant function based on tail currents

The three models introduced above are given in their ‘wild-type’ formulation. The ‘mutant’ formulation has only a single difference, a modified INa current, as described in the following and illustrated in Fig. 1.

Figure 1 Inactivation time constant as a function of membrane potential.

(A) Voltage-dependent time constant for mutation τh* and wild-type (τh). Insets show the response of h to a voltage-clamp protocol. (B) shows the deinactivation (i.e., recovery from inactivation) process as a response to a step in voltage from −10 mV to −120 mV. (C) shows the inactivation process by stepping the voltage from −120 mV to −10 mV. The intersections of the h-curve with the 1/e- and (1−1/e)-lines, respectively, show the actual time constants. For deinactivation τh* is three-fold smaller than τh. For inactivation τh* is three-fold lager than τh.

From experimental data we know that the mutation leads to a two- to four-fold faster deinactivation (Dichgans et al., 2005) and to a two- to four-fold slower inactivation (Vanmolkot et al., 2007). We checked the robustness of our simulations within this range. The simulations presented here, however, were performed at an intermediate value of a three-fold change.

To change the responsiveness of inactivation and deinactivation accordingly, we need to modify the time constant τh of the gating variable h. In the mutant model this time constant is replaced by (42) τh∗V=τhV⋅κ1⋅tanhσ⋅V−Vmax+κ2.

The parameter Vmax shifts the sigmoidal tanh-function to the position of the maximum of the time constant function τh(V). The slope factor of the sigmoidal tanh-function is σ = 0.1 to ensure sufficiently rapid convergence to the limit of a three-fold change. The other parameters are κ1 = 1.335 and κ2 = 1.665. These parameters result from the two constraints κ1 + κ2 = f and κ2−κ1 = 1/f for an f-fold change. We chose f = 3.

To test the mutant time constant τh∗, we simulated the experimental protocol performed by Dichgans et al. (2005) in the computational model. The membrane voltage is clamped to a holding potential of −120 mV and then stepped to a potential of −10 mV. At −120 mV the h-gate is completely deinactivated, i.e., open. The step to −10 mV causes the h-gate to inactivate. Therefore, we can measure the time constant of inactivation with this protocol (see Fig. 1B). In contrast, holding the membrane potential at −10 mV and then stepping back to −120 mV allows us to measure the time constant of the process of deinactivation. At −10 mV the h-gate is completely inactivated, i.e., closed, and the step to −120 mV causes the gate to deinactivate again, i.e., the gate reopens. An illustration of this protocol can be found in Fig. 1C. By using this procedure and measuring the two different time constants, it was assured that the chosen parameters lead to a 3-fold slower inactivation and a 3-fold faster deinactivation. The main part of Fig. 1 shows the inactivation time constants τh (black line) and τh∗ (green line) for the wild-type and mutant model, respectively, as a function of the membrane potential V.

Note that in the Hodgkin–Huxley formalism, the gating subunits of a channel are assumed to be identical and the inactivation and deinactivation as being independent. Therefore this formalism cannot represent certain dependencies in a straightforward manner in the kinetic states. For example, the inactivation of the Na+ channel (represented by the h-subunit) has a greater probability of occurring when all subunits are open, therefore the inactivation depends on activation (represented by the three m-subunits). This violates the assumption of independent gating. Because of this independence in the HH formulation, the dynamics of the h-gate is only described by a single time constant function τh. An alternative ansatz is to use a Markov model to compute the occupancy of the channel in its various kinetic states as done by Clancy & Rudy (1999).

Results

Three different models are investigated, a model of action potentials (AP), a model of spreading depression (SD), and a model of anoxic depolarization (AD). These models describe normal cell functions in terms of the dynamic repertoire either without genetic defect (three wild-type models) or with altered cell functions in FHM3 (three mutant models). The three mutant models (AP, SD, and AD) are the same as the wild-type models except that the INa current has a different voltage-gating mechanism in the fast gating variable h. This is described in the wild-type model by the time constant τh and in the mutant model by τh∗ (see Methods). The observed functional consequences of FHM3 occur on time scales ranging from milliseconds to several tens of seconds.

Mutant AP with marked plateau, increased responsiveness, delayed excitation block, and firing onset unchanged

We first consider the shape of APs. The AP is rather directly affected by FHM3 through altered voltage gating in h. In other words, the results are consistent with the measured tail currents and therefore the results for a mutant AP are even to some degree predictable. This situation will change, when we model dynamics separated three orders of magnitude from AP dynamics.

For a single AP stimulated by a transient applied current Iapp(t) of 3 ms duration and 3 µA cm−2 amplitude (labeled ‘excitatory’ in Fig. 2), we observe that the mutant model compared to wild-type model leads to a prolonged AP with a marked plateau. This is consistent with the larger inactivation time constant τh∗Vdep of the mutant as compared to the wild-type inactivation time scale τh(Vdep), cf. tail currents in Fig. 1B. Note that we omitted before the explicit voltage dependency of the time constants, but now we make the dependency explicit because the mutant time constant function τh∗V is in FHM3 increased only for the regime of the membrane potential V being depolarised. This voltage regime is indicated by the superscript “dep” and it corresponds to an inactivation of h (closed h gates).

Figure 2 Spiking model.

Comparison of wild-type (A) and mutant (D) spiking behavior. The main plots show bifurcation diagrams by varying the external current Iapp. For the wild-type model Hopf bifurcations can be found at Iapp = 9.78 µA cm−2 and Iapp = 154.52 µA cm−2. For the mutant model Hopf bifurcations occur at Iapp = 9.72 µA cm−2 and Iapp = 175.02 µA cm−2. (B) and (E) show behavior in the oscillatory regime as a response to a constant input current of 12 µA cm−2. (C) and (F) show the response of the models in the excitatory regime to a 3 ms long current pulse with amplitude 3 µA cm−2.

Furthermore and a bit more subtle to observe, the mutant dynamics reacts faster to a sudden brief stimulation. The mutant model fires an AP that reaches its maximal amplitude just below 2 ms after the Iapp is turned off again, while in the wild-type model the maximal amplitude is reached only after about 3 ms. Again, this is also consistent with the defect in the time constant function τh∗V. In this case it is explained by the decreased and therefore faster regime τh(Vpol) compared to the wild-type. The mutant time constant function τh∗V is decreased for V being in the polarised resting state indicated by the superscript “pol”, cf. tail currents in Fig. 1A. This is the regime of deinactivation (open h gates).

The modified AP profile is also observed during spiking, i.e., in the oscillatory regime, when a constant Iapp larger than—by definition (see below)—the rheobase current Irh is applied. Individual APs in the spike train show this plateau (labeled ‘oscillatory’ in Fig. 2). As a result the spiking frequency is reduced in the mutant model, despite the overall increased responsiveness (Fig. 3). This decreased spiking frequency can be associated with hypoexcitability as the neural response is usually characterized by the firing-rate function.

Figure 3 Nonlinear firing-rate function F(Iapp) for wild-type model (black, solid) and mutant model (green, dashed).

To get some further quantitative measures of the effects of FHM3 with regard to excitability, we investigated the change of stability in the resting state by varying the input current Iapp. This is a bifurcation analysis (Fig. 2). The determined two so-called bifurcation points mark the beginning and end of the oscillatory spiking regime. The first Hopf bifurcation point (HB1) is the onset of oscillation at a minimal value of Iapp, which is the definition of the rheobase current Irh. For the wild-type model the first Hopf bifurcation (HB1) is at IappHB1≡Irh=9.78µA cm−2 and the second Hopf bifurcation (HB2) at IappHB2=154.5µA cm−2, which determines the excitation block as the oscillation ceases at this point. For the mutant model these Hopf bifurcations occur at Irh = 9.72 µA cm−2 and IappHB2=175.0µA cm−2. The first Hopf bifurcations (HB1) are subcritical, while HB2 are both supercritical. This means that if the Iapp is not slowly ramped towards the rheobase current Irh, one can observe the oscillatory regime even before the Irh. Hence the two firing-rate functions in Fig. 3 start slightly before the values given here for HB1, with the mutant model starting again earlier.

With regard to the rheobase current, the values for the wild-type and mutant differ by less then 0.6%, with the mutant value being smaller, which, at least in principal, corresponds to hyperexcitability, though due to the small magnitude this seems negligible for all practical purposes. However, the excitation block observed at the second critical transition HB2 occurs at larger values of Iapp for the mutant model. The mutant channels tolerate an increased maximal IappHB2 by 13% compared to the wild-type. This means that the mutant neurons exhibit oscillatory behavior in a larger range of applied currents. Therefore, this shift establishes a gain-of-function, which indicates hyperexcitability.

To summarize, while the reduced firing frequency indicates hypoexcitability, increased responsiveness and delayed excitation block indicate hyperexcitability.

Mutant more vulnerable to SD

We now focus on effects of FHM3 upon cellular functioning that occurs in the same neural substrate that generates APs but on time scales at least three orders of magnitude separated from AP dynamics, that is, effects that occur during several tens of seconds up to minutes. This is the time scale of SD. It is therefore relevant for pathological conditions, for instance, in migraine with aura. In accordance with this pathophysiological context, we select the stimulations of SD in the wild-type and mutant model as rather large perturbations to neural homeostasis such as a compromised energy supply during focal hypoperfusion that induces and occurs in conjunction with migraine aura symptoms (Olesen et al., 1993; Friberg et al., 1994).

In particular, we investigate the effect of a breakdown of the Na+-K+-pump upon the membrane potential V and reversal potentials ENa and EK. For this purpose the maximum pump rate ρ is linearly down-regulated to 20% of its physiological value within 10 s, then ρ is kept at 20% for a variable time window, and finally ρ is linearly up-regulated back to 100% within 5 s. The stimulation trace of ρ is shown in Fig. 4 with the dashed-dotted line. The specific choice of the variable time window is additionally marked for the wild-type and mutant stimulation trace by an annotated two-headed arrow. Let us remark that in our studies we also used two other perturbations, namely a transient increase in extracellular K+ concentration (by increasing Kbath) and a large current pulse Iapp, with basically the same results (not shown).

Figure 4 Spreading depression model.

Development of SD in wild-type (A) and mutant (B) models. A SD is elicited by down-regulating the pump current to 20% of its maximal value for 13.6 s (wild-type) and 7.2 s (mutant), respectively (see blacked dashed line). The red and blue dashed lines show the temporal development of the sodium and potassium reversal potentials.

We determined the minimal duration of the variable time window with reduced pump rate (20%) that is just no longer tolerated and results in a long lasting but transient breakdown of the reversal potentials ENa and EK characteristic for SD. For this purpose we increased the variable time window by 0.1 s steps. While the wild-type model could not tolerate a period of 13.6 s of reduced pump rate at 20%, the mutant model was less robust and could not tolerate a period of 7.2 s of reduced pump activity (Fig. 4). Therefore, the mutant model is approximately only half (53%) as robust to periods of reduced ion pump activity as the wild-type model is.

Shorter stimulation periods did not lead to full blown SD signals. In this case, the spiking ceased about a second after the interval began that increased the pump rate back from 20% to 100% (this interval lasts 5 s) and, more importantly, both membrane potential V and reversal potentials ENa and EK recovered within only a few seconds back to physiological values (not shown). Thus, SD profiles of these potentials, which followed longer stimulation periods, are clearly distinguished by a all-or-none phenomenon. Not only do membrane potential V and reversal potentials ENa and EK change dramatically after the stimulation is off, but also full recovery from SD to the initial physiological values takes very long. Of course recovery reaches the resting state only asymptotically. For up to one to two hours the changes in particular in ENa are observable, while the signals in Fig. 4 are shown only for 100 s. It is noteworthy that the neuronal state is already back to basic functioning emitting APs if stimulated after the repolarization, that is, even if the resting state is not fully recovered. Similar dynamics is described in other computational models of SD by Kager, Wadman & Somjen (2000), Yao, Huang & Miura (2011) and Hübel, Schöll & Dahlem (2014).

To summarize, in terms of susceptibility to SD the mutant model is hyperexcitable. This seems to be in contrast to the major effect of the mutant upon the AP firing frequency that indicates that the mutant model is hypoexcitable (Fig. 3). This will be further discussed in the Discussion.

Effects in anoxia model consistent with SD model

Last, we study a model of AD (Zandt et al., 2011); the AD model shares many features with the SD model but is more detailed (see Methods) and hence effects obtained with this model serve as control to compare them with effects obtained from the SD model. The model was first published to study slow waves after decapitation in a computational model (Zandt et al., 2011). By repeating this with a mutant version of this model, our focus is set very similar to the previous section. In the decapitation study, anoxia is modeled by completely switching off all pump, glial, and diffusion currents, see Fig. 5. In fact, Fig. 5A with the wild-type model is a reproduction of the simulations performed by Zandt et al. (2011).

Figure 5 Anoxia model.

Response of wild-type (A) and mutant (B) membrane potential to a complete breakdown of pump, glial and diffusion currents at t = 5 s (black dashed vertical line). Red and blue dotted lines show the Na+ and K+ reversal potentials over time. The time from the onset of spiking until the beginning of the excitation block is approximately 6.7 s without and 2.7 s with mutation.

Note that patients with migraine with aura are at greater risk for stroke (Kurth & Diener, 2012). Thus there is a rationale to perform this comparison beyond the mere confirmation of plausibility of our results obtained above with the SD model. However, the multiplicity of potential links include not only common genetic risk factors but also indirect links like common triggers outside the brain, e.g., microemboli caused by cardiac shunts. Furthermore, the model investigated by Zandt et al. (2011) is derived from a model suggested by Cressman et al. (2009). This model exhibits periodic bursting similar to seizure activity. Both migraine and epilepsy have genetically based forms caused by various mutations in genes, while the mutation in FHM3 differs markedly within the several mutations in SCN1A therein that it is not associated with epilepsy (see Introduction). Investigating the underlying ion homeostasis in the three conditions of epilepsy, migraine, and stroke may yield interesting results in future investigations of computational models that can unify certain dynamical aspects and link disease genotype to phenotype. However, this is clearly beyond the scope of this study.

In this study, let us only refer to the dynamics resulting from switching off all pump, glial, and diffusion currents until the excitation block and compare the wild-type and mutant model. After a gradual rise of the membrane potential that lasts in either case about 30 s (note that the simulated ‘decapitation’ occurs in Fig. 5 at t = 5 s), the membrane potential reaches the AP threshold, subsequently resulting in a final burst of spiking. These initial, less than a minute lasting, phases in the wild-type and mutant model are indeed very similar to the initial phases in the SD model following a transient energy failure. A minor difference is that the gradual rise is overall slower, but this is explained by a slightly different geometry (larger extracellular space) and by the chloride ion dynamics (Hübel, Schöll & Dahlem, 2014). The similarity supports the robustness of our results, as this model is an established model showing anoxia (Zandt et al., 2011) and seizure activity (Cressman et al., 2009; Cressman et al., 2011).

To summarize, also for AD the slow gradual fall of the potentials does not significantly differ during the initial leak phase in the wild-type and mutant model, while once the model is spiking the excitation block occurs about 2.5-times faster, corresponding to a faster breakdown of ion gradients due to spiking, in the mutant model.

Discussion

Our main result is that the mutant model is more susceptible to spreading depression (SD). With our computational model, we bridge the gap between the tail currents measured by Dichgans et al. (2005) and altered cell function that constitutes the phenotype of migraine with aura. Importantly, in a computational model we can follow in all needed detail how the complex interactions of channel dynamics lead to altered cell function. A similar approach was taken, for instance, to link a genetic defect to its cellular phenotype in a cardiac arrhythmia by Clancy & Rudy (1999).

In the discussion, we mainly highlight aspects of hypoexcitable vs. hyperexcitable and the concept of a threshold.

Hypoexcitable vs. hyperexcitable

The increased susceptiblility to SD does not contradict the reduced firing frequency for a given stimulation current Iapp, although this change in firing frequency indicates that the mutant model is hypoexcitable.

Firing a single action potential (AP) is a form of cellular excitability manifested as a transmembrane voltage jump without significant changes in ion concentrations. SD is a form of cellular excitability manifested by massive changes in ion concentrations. There is not necessarily a direct relation between the two excitable systems, not even with regard to merely classifying terms such as hypoexcitable and hyperexcitable. Rather, AP and SD can be viewed as largely independent phenomena, because while sharing the same neural substrate, AP and SD are separated by time scales differing in three orders of magnitude (see below). Notwithstanding, the massive breakdown of ion gradients in SD is, of course, mediated by APs that occur on the fast time scale.

In our view, “hypoexcitable” vs. “hyperexcitable” is in any case too simple a classification scheme even considering AP and SD in isolation on their respective time scale. To support this criticism of classifying neural dynamics in migraine, let us mention that this problem was also addressed in the psychophysical and clinical contexts, see studies by Shepherd (2001) and Coppola, Pierelli & Schoenen (2007) and references therein; further support comes from the mathematical picture (below)—which are two sides of the same coin.

To illustrate this with only a single example, consider, as already mentioned above, that the mutant channels exhibit an increased range of spiking activity with a delayed excitation block by 13% compared to the wild-type. We argued that this larger spiking range establishes a gain-of-function. Consider further the increased responsiveness of the mutant model. Both indicate a form of hyperexcitability with regard to AP. In contrast, the change in firing frequency of AP indicates at the same time that the mutant model is hypoexcitable (Fig. 3).

SD susceptibility

How do these three diverse effects observed for APs (delayed excitation block, increased responsiveness, and lower firing frequency) manifest on the longer time scale under the condition of SD?

In terms of susceptibility to SD, the shifted excitation block (see HB2 in Fig. 2) might misleadingly suggest that the mutant model is less susceptible to SD. This is similar to the lower firing frequency that we considered above. Since the characteristic sustained breakdown of the reversal potentials ENa and EK is ignited in our model only if the system is driven by any stimulation into the excitation block, its delay in the mutant model seems to suggest that a longer stimulation may be needed and therefore a higher threshold exists.

To show the actual situation in Fig. 4, we highlighted a critical time window by a gray shade. This critical time window opens with start of the reduced pump rate recovery (from 20% back to 100%) and it closes with the beginning of the excitation block. Considering only the delay of the excitation block and the low frequency, it may seem surprising at first, this critical period lasts 3.4 s in the wild-type model and only 2.5 s in the mutant model. Note that this ‘paradox’ can also be observed in the overall shorter duration of the whole initial firing pattern in the mutant SD model. Our attention should be on signals that can actually be measured in a clinical setting, hence our focus is on these signals also in the presentation of the computational model, where we can “measure” everything. The reduced pump rate corresponds to hypoperfusion signals. The excitation block in SD corresponds to the first peak in an electroencephalography (EEG) signal, cf. the work by Zandt et al. (2011) where the simulated membrane potential is averaged and high-pass filtered, cut-off at 0.1 Hz, to estimate the EEG—although this EEG might only be observable intracranially.

That the mutant model is more susceptible to spreading depression (SD) is exclusively explained by the much larger amount of ions transferred across the membrane during spiking. This, in particular the intracellular ion concentration, cannot easily be measured even in an in vitro setup. In Fig. 4, we see this by the much steeper slope of the reversal potential EK in the mutant model.

The multidimensional concept of thresholds

The complex question about susceptibility requires a deeper understanding of what a threshold is. In fact, the very reason why we have to get beyond the idea of “hypoexcitable” vs. “hyperexcitable” as a useful characterization of the system (see above) is that there is no one-dimensional ansatz to determine a threshold as a demarcation.

Before explaining this further, let us give one more explicit example. In other model variants of SD (Kager, Wadman & Somjen, 2000), a stimulation of SD may even stop before the excitation block is reached. In this case a sustained afterdischarge carries the system into the depolarisation block that then marks the start of the actual SD events. Clearly, in this case the depolarisation block cannot be considered being the actual threshold, because the system is ‘before’ this point when the stimulation is already off again.

In general, excitability or all-or-none phenomena do not possess a threshold in terms of single quantity, whether it is a particular membrane depolarisation that demarcates the all-or-none response in the form of an AP or a critical duration of hypoperfusion that demarcates the all-or-none response in form of SD. A detailed analysis of neural models shows that a threshold is a multidimensional surface (manifold) not a single number as first shown by FitzHugh (1955) and as discussed in a modern style by Mitry et al. (2013) and applied to migraine by Dahlem (2013). So the actual use of computational models goes far beyond numerical simulations. We gain a deeper understanding of the principal mechanisms in precise mathematical relationships, of which we can only give a very general overview in this paper.

Additional Information and Declarations

Competing Interests

Author Contributions

The authors declare there are no competing interests.

Markus A. Dahlem conceived and designed the experiments, performed the experiments, analyzed the data, contributed reagents/materials/analysis tools, wrote the paper, prepared figures and/or tables, reviewed drafts of the paper.

Julia Schumacher performed the experiments, analyzed the data, wrote the paper, prepared figures and/or tables, reviewed drafts of the paper.

Niklas Hübel performed the experiments, analyzed the data, wrote the paper, contributed reagents/materials/analysis tools, reviewed drafts of the paper.

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
