# Peer review of "Linking a genetic defect in migraine to spreading depression in a computational model"

_PeerJ, doi:10.7717/peerj.379_

## Round 0.1 · original submission · Minor Revisions

Thank you for submitting your manuscript to PeerJ. The reviewers and I concur that the manuscript is suitable for publication pending the required revisions.

·

Basic reporting

This is a well-written and clear manuscript that computationally examines the consequences of a genetic mutation of the sodium channel. Three different models are used to address the effects at the level of the action potential, the more elaborate spreading depression signal, and anoxic depolarization. The results are a useful contribution and point out that the effects are manifested in unexpected and perhaps counterintuitive ways. The results inform the important topics addressed in the discussion regarding the ambiguity of common used terminology.

Two minor “basic reporting” points:
In the discussion of figure 2, the phrase “These Hopf bifurcations are subcritical” appears. The wording surrounding this makes it unclear which bifurcations are being referred to. I would expect both the wild-type and mutant versions of HB1 to be subcritical, and both HB2 bifurcations to be supercritical.

In the caption to Figure 1, the description of the right inset says that the voltage steps from -10mV to -120mV. It should be the other way around. Also, there is no direct reference in the text to the main part of Figure 1. Describing this when it is introduced would help make the discussion of the “marked plateau” on page 8 easier to understand.

Experimental design

No concerns. Although the work is computational in nature, it is fully appropriate for PeerJ.

Validity of the findings

No concerns.

Additional comments

Please note that the first paper listed in the references section was published in 2011, not 2010. Also, the authors should include a citation to the erratum to Cressman et al. (i.e., JCNS 30, 781 (2011)).

A few grammar/wording suggestions (these do not affect the clarity of the manscript):
“Prolonged” is misspelled as “prolognoed”. “In form of” should be “in the form of”. “Boththe” needs a space. In many instances the word “rational” should be replaced with “rationale”. After Eq. (42), the text should read “These parameters result from the two constraints…”. “Occur” on page 10 should be “occurs”. P. 13: “higher threshold” is better than “larger threshold”.

Reviewer 2 ·

Basic reporting

The article “Linking a genetic defect in migraine to spreading depression in a computational model” by Dahlem et al., presents a computational study of the effect of Familial hemiplegic migraine mutant (FHM3) on a single neuron behavior. Specifically, authors present 3 models for: (1) action potentials (simple spiking at ms scale) involving only Hodgkin-Huxley dynamics (HH), (2) spreading depression, and (3) anoxic depolarization (at tens of seconds) combining HH dynamics with dynamic ion concentrations. Each model then has 2-versions one each for wild-type and FHM3 mutation. The study then shows how the dynamics changes at these different temporal scales in the mutant case as compared wild-type cell. The most interesting result of the study is that even the frequency of mutant cell is smaller as compared to control cell, the mutant model is more susceptible to spreading depression. Based on their results, the authors then argue that “hypoexcitability” and “hyperexcitability” is too simple a classification for defining neuronal behavior. Over all I found the model biophysically relevant; this paper very interesting, clearly written, and suitable for publication by PeerJ.

I have a few minor comments.

(1) The membrane surface, membrane volume, and extracellular space volume seems to be too small (may be a unit error?).
(2) The units in Figure 2 legend need to be corrected.
(3) The reference to Figure 2 on page 9 (first paragraph) should be corrected, i.e. the “oscillatory” label is in Fig. 2 not Fig. 1.

Experimental design

See basic reporting

Validity of the findings

See basic reporting

Additional comments

see basic reporting

---

## Round 0.2 · accepted · Accept

The revised version successfully covers all the raised issues.